# Association between weight-adjusted waist index and periodontitis: A population-based study

Lihan Xu[1], Yuntao Liu[1], Chenchen Zhao[1], Jiaying Wang[1], Haojing Zhou[2]*

1 College of Stomatology, Hangzhou Normal University, Hangzhou, Zhejiang Province, China, 2 Zhejiang Chinese Medical University, Hangzhou, Zhejiang Province, China

* haojingz0826@126.com

## Abstract

### Objective

This study aims to examine the association between the Weight-adjusted Waist Circumference Index (WWI) and the prevalence of periodontitis, providing novel evidence on the link between central obesity and periodontal health.

### Methods

A cross-sectional study was conducted with 10,289 participants enrolled from NHANES 2009 to 2014. WWI was calculated by dividing waist circumference by the square root of weight. We employed a multivariate logistic regression model and smoothed curve fitting method to evaluate the relationship between WWI and periodontitis. We also compared different subgroups and analyzed the interaction effects.

### Results

A significant positive association between WWI and periodontitis was observed in 10,289 participants aged ≥30 (OR: 1.20, 95% CI: 1.12–1.28). Upon categorizing WWI into quartiles, the top quartile group exhibited a 27% increased prevalence of periodontitis compared to the bottom quartile (OR: 1.27, 95% CI: 1.10–1.46; P for trend = 0.001). Among individuals aged 30 to 60, the strength of this positive correlation is more pronounced than in those aged 60 and above.

### Conclusions

WWI demonstrates a positive correlation with periodontitis with a particularly pronounced impact on moderate periodontitis, suggesting its potential to improve periodontitis prevention in a broad population.

**Data Availability Statement:** All relevant data are within the manuscript and its Supporting Information files.

**Funding:** The author(s) received no specific funding for this work.

**Competing interests:** The authors have declared that no competing interests exist.

## Introduction

Periodontitis, a chronic inflammatory disease, arises from the interaction of pathogenic bacteria and the host immune response, which gradually erodes the periodontal supporting structure and ultimately results in tooth loss [1]. Worldwide, periodontitis affects approximately 45–50% of the population, with the severe case impacting 11.2%, ranking it as the sixth most common disease globally [2]. In the U.S., individuals over 30 experience a prevalence rate of 42.2% for periodontal disease. Additionally, periodontitis has been found to have associations with numerous systemic disorders, such as diabetes [3], cardiovascular diseases [4], and Alzheimer's disease [5], thereby imposing substantial health and economic burdens on society.

Obesity is defined as improper or excessive adipose tissue buildup [6]. As awareness of the role of adipocytes in inflammation and immune regulation gains, obesity is increasingly seen as a chronic systemic condition that can impact the onset and progression of periodontal disease [7, 8]. Traditional obesity assessment tools like Body Mass Index (BMI) and Waist Circumference (WC) are utilized yet have drawbacks [9, 10]. While the link between BMI, WC, and periodontitis has been reported in various studies, it is still controversial [11–14]. According to research, BMI fails to distinguish between central and peripheral adiposity and varies with age, gender, and ethnicity [15, 16]. Park et al. introduced the Weight-adjusted Waist Index (WWI) in 2018, a brand-new indicator that refines waist circumference measurement by reducing its correlation with BMI, thus offering a more precise assessment of central obesity [17]. Using this new indicator to explore the relationship between periodontitis and obesity might further reveal a more precise relationship between them, which might aid in periodontitis prediction.

Nevertheless, the correlation between WWI and periodontitis has not been thoroughly investigated. Therefore, we seek to assess the possible link between WWI and periodontitis prevalence, utilizing statistics from the National Health and Nutrition Examination Survey (NHANES) spanning 2009–2014.

## Materials and methods

This cross-sectional study followed the Strengthening the Reporting of Observational Studies in Epidemiology (STROBE) statement (S1 Table) [18].

### Data collection

This cross-sectional study utilized data from NHANES spanning 2009–2014. This extensive survey concentrates on evaluating the health and nutritional status of the U.S. populace [19]. Extensive information on the NHANES survey design is accessible at https://www.cdc.gov/nchs/nhanes/index.htm.

From the NHANES 2009–2014 cohort, 30,468 individuals were initially considered. Exclusions were made for those lacking data on periodontitis (19,804 individuals) and WWI (375 individuals in total; including 321 without waist circumference data and 54 without weight data). Ultimately, the study's sample consisted of 10,289 participants, as detailed in Fig 1.

### Evaluation of WWI

The WWI is formulated for obesity assessment, which was obtained by dividing the WC (in cm) by the square root of body weight (in kg) [20]. In our analysis, WWI was utilized as an exposure variable. We considered WWI as a continuous metric and divided the study subjects into quartiles according to their WWI scores for a more detailed examination.

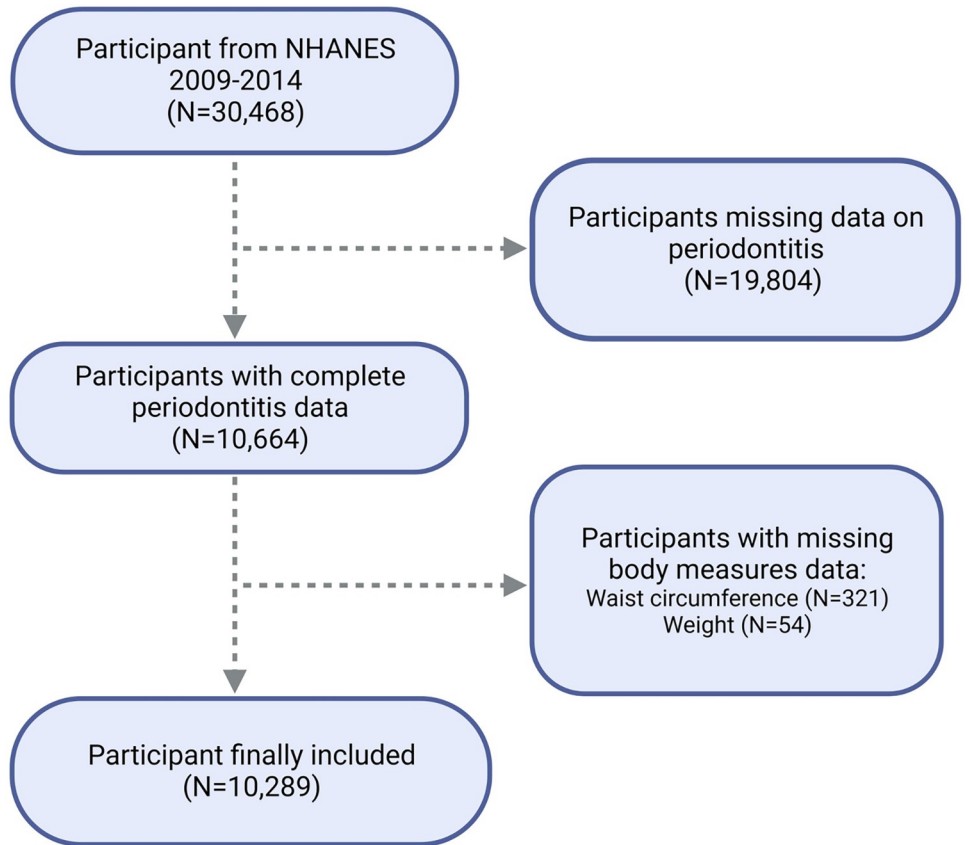

**Fig 1. Flowchart of population included in our final analysis.**

### Evaluation of periodontitis

Periodontal examination includes measuring both Clinical Attachment Loss (CAL) and Probing Pocket Depth (PPD) at six locations per tooth excluding third molars, using the FMPE protocol [21]. According to the consensus of the Centers for Disease Control and Prevention and American Academy of Periodontology, periodontitis was categorized into four stages: no, mild, moderate, or severe periodontitis [22]. The criteria for each stage are as follows: (1)Mild periodontitis: more than 2 interdental locations with CAL $\geq$3 mm but <4 mm, more than 2 interdental locations with PPD $\geq$4 mm but <5 mm (on different tooth), or a single location with PPD $\geq$5 mm; (2)Moderate periodontitis: more than 2 interdental locations with CAL $\geq$4 mm but <6 mm or more than 2 interdental locations with PPD $\geq$5 mm (on different tooth); (3)Severe periodontitis: more than 2 interdental locations with CAL $\geq$6 mm (on different tooth) and at least one interdental location with PPD $\geq$5 mm; (4) Participants not meeting any of these criteria were considered to have no periodontitis [23]. Many similar studies have used this criterion to define periodontitis [24–26].

### Selection of covariates

To ensure highly accurate results, we investigated several sociodemographic and behavioral factors, as well as systemic diseases linked to periodontitis, as potential confounders. The included covariates were: age (years), gender (male/female), racial/ethnic identity (Non-Hispanic White, Non-Hispanic Black, Mexican American, or Other race/Multiracial), educational

background (below high school, high school graduate, or over high school), Poverty Income Ratio (PIR), smoking condition (indicating if the participant has smoked at least 100 cigarettes in their life: Yes/No), alcohol consumption frequency over the past year, diabetes (yes/no), levels of Triglycerides, and High-density lipoprotein cholesterol.

## Statistical analysis

Our descriptive analysis assessed the demographic characteristics of participants across various WWI quartiles using chi-square and t-tests. We investigated the relationships between WWI and periodontitis, including its varying degrees of severity, employing multivariate linear and logistic regression models. To assess robustness, we examined linear trends between WWI and periodontitis by conducting a trend test after categorizing WWI as a quartile variable instead of a continuous one. The analysis was structured into three distinct models. Model 1 was conducted without any covariate adjustments. Model 2 incorporated adjustments for age, gender, and race. Model 3, the most comprehensive, included adjustments for all the covariates under consideration. Additionally, we utilized smoothed curve fitting to explore the possible non-linear connection between WWI and periodontitis. The threshold effect within this relationship was evaluated using two distinct linear regression models. Subgroup analysis was carried out, focusing on the WWI-periodontitis relationship across individuals of different genders, ages, races, educational levels, drinking behaviors, and diabetes. Interaction tests were further applied to assess the uniformity of these associations across the different subgroups. All statistical analysis were executed using R software (version 4.2) and EmpowerStats (version 5.0), considering a two-sided p-value of less than 0.05 as statistically significant.

## Results

### Participants' baseline characteristics

Our study included 10,289 individuals, with an average age of 51.84 ± 14.17 years; 49.51% were male, and 43.02% identified as non-Hispanic white (Table 1). All individuals had a mean WWI of 11.08 ± 0.77. The average WWI values for each quartile (Q1-Q4) were 8.62–10.55, 10.55–11.07, 11.07–11.60, and 11.60–14.79, respectively. The overall prevalence of periodontitis was 50.6%, showing an upward trend in line with increasing WWI quartiles (Quartile 1: 41.17%, Quartile 2: 48.13%, Quartile 3: 53.58%, Quartile 4: 59.50%; P < 0.0001). Individuals in the upper WWI quartiles exhibited characteristics such as advanced age, a higher likelihood of being female, and lower incomes and education levels. Additionally, they exhibited a greater prevalence of diabetes and moderate-to-severe periodontitis, higher alcohol intake, elevated triglyceride levels, and reduced HDL cholesterol levels.

### Association between WWI and periodontitis

Our analysis employed three models to examine the relationship between the WWI and the prevalence of total periodontitis (Table 2). We identified a substantial positive relationship between WWI and periodontitis when considering WWI as a continuous variable, which remained statistically significant across all three models. Even after eliminating key demographic and other disease-related variables in Model 3, this association remained consistently significant. In the model with full adjustments, we observed that with every unit rise in WWI, there was a corresponding 20% increase in the periodontitis prevalence (Model 3, OR: 1.20, 95% CI: 1.12–1.28). We subsequently divided WWI into quartiles and observed that this significant correlation persisted among all three models. In the unadjusted model (Model 1), higher WWI groups tend to exhibit an increasing prevalence of periodontitis compared to the lowest

**Table 1. Basic characteristics of participants by weight-adjusted waist index quartile.**

| Characteristics | Weight-adjusted-waist index quartile (cm/√kg) | | | | Overall | P-value |
|---|---|---|---|---|---|---|
| | Q1 (8.62–10.55) | Q2 (10.55–11.07) | Q3 (11.07–11.60) | Q4 (11.60–14.79) | | |
| | N = 2572 | N = 2572 | N = 2572 | N = 2573 | | |
| Age (years) | 44.80 ± 11.66 | 49.92 ± 13.05 | 54.08 ± 13.74 | 58.55 ± 14.30 | 51.84 ± 14.17 | <0.001 |
| Gender, (%) | | | | | | <0.001 |
| Male | 1558 (60.58%) | 1424 (55.37%) | 1227 (47.71%) | 885 (34.40%) | 5094 (49.51%) | |
| Female | 1014 (39.42%) | 1148 (44.63%) | 1345 (52.29%) | 1688 (65.60%) | 5195 (50.49%) | |
| Race/ethnicity,(%) | | | | | | <0.001 |
| Non-Hispanic White | 1147 (44.60%) | 1096 (42.61%) | 1051 (40.86%) | 1132 (44.00%) | 4426 (43.02%) | |
| Non-Hispanic Black | 704 (27.37%) | 499 (19.40%) | 481 (18.70%) | 423 (16.44%) | 2107 (20.48%) | |
| Mexican American | 181 (7.03%) | 356 (13.85%) | 445 (17.31%) | 501 (19.47%) | 1483 (14.41%) | |
| Other race/multiracial | 540 (21.00%) | 621 (24.14%) | 595 (23.13%) | 517 (20.09%) | 2273 (22.09%) | |
| Education level, (%) | | | | | | <0.001 |
| Less than high school | 354 (13.77%) | 563 (21.92%) | 665 (25.87%) | 811 (31.57%) | 2393 (23.28%) | |
| High school | 538 (20.93%) | 500 (19.47%) | 584 (22.71%) | 575 (22.38%) | 2197 (21.38%) | |
| More than high school | 1678 (65.29%) | 1505 (58.61%) | 1322 (51.42%) | 1183 (46.05%) | 5688 (55.34%) | |
| PIR | 2.94 ± 1.64 | 2.74 ± 1.59 | 2.58 ± 1.59 | 2.28 ± 1.46 | 2.63 ± 1.59 | <0.001 |
| Smoked at least 100 cigarettes in life | | | | | | 0.536 |
| Yes | 1097 (42.68%) | 1139 (44.28%) | 1142 (44.42%) | 1141 (44.35%) | 4519 (43.93%) | |
| No | 1473 (57.32%) | 1433 (55.72%) | 1429 (55.58%) | 1432 (55.65%) | 5767 (56.07%) | |
| How often drink alcohol over the past 12 months | 2.51 ± 2.06 | 3.29 ± 27.87 | 3.09 ± 24.96 | 3.32 ± 31.74 | 3.05 ± 24.55 | 0.002 |
| Diabetes, (%) | | | | | | <0.001 |
| Yes | 117 (4.55%) | 268 (10.43%) | 460 (17.88%) | 688 (26.76%) | 1533 (14.91%) | |
| No | 2454 (95.45%) | 2301 (89.57%) | 2112 (82.12%) | 1883 (73.24%) | 8750 (85.09%) | |
| Triglycerides (mg/dL) | 194.32 ± 37.00 | 199.32 ± 40.49 | 198.89 ± 41.52 | 198.00 ± 44.09 | 197.63 ± 40.90 | <0.001 |
| High-density lipoprotein cholesterol (mg/dL) | 57.03 ± 17.22 | 53.20 ± 16.12 | 50.75 ± 14.66 | 50.09 ± 13.94 | 52.77 ± 15.77 | <0.001 |
| Periodontitis, (%) | | | | | | <0.001 |
| No | 1513 (58.83%) | 1334 (51.87%) | 1194 (46.42%) | 1042 (40.50%) | 5083 (49.40%) | |
| Mild | 143 (5.56%) | 127 (4.94%) | 115 (4.47%) | 112 (4.35%) | 497 (4.83%) | |
| Moderate | 693 (26.94%) | 832 (32.35%) | 996 (38.72%) | 1148 (44.62%) | 3669 (35.66%) | |
| Severe | 223 (8.67%) | 279 (10.85%) | 267 (10.38%) | 271 (10.53%) | 1040 (10.11%) | |

Mean ± SD for continuous variables: the P value was calculated by the linear regression model. (%) for categorical variables: the P value was calculated by the chi-square test. Abbreviation: Q, quartile; PIR, Ratio of family income to poverty.

**Table 2. The associations between weight-adjusted waist index and periodontitis.**

| Exposure | Model 1 [OR (95% CI)] | Model 2 [OR (95% CI)] | Model 3 [OR (95% CI)] |
|---|---|---|---|
| WWI (continuous) | 1.48 (1.40, 1.56) | 1.38 (1.30, 1.47) | 1.20 (1.12, 1.28) |
| WWI (quartile) | | | |
| Quartile 1 | reference | reference | reference |
| Quartile 2 | 1.33 (1.19, 1.48) | 1.18 (1.05, 1.33) | 1.06 (0.94, 1.21) |
| Quartile 3 | 1.65 (1.48, 1.84) | 1.36 (1.20, 1.54) | 1.13 (0.99, 1.29) |
| Quartile 4 | 2.10 (1.88, 2.35) | 1.73 (1.51, 1.97) | 1.27 (1.10, 1.46) |
| P for trend | < 0.001 | < 0.001 | 0.001 |

Model 1: no covariates were adjusted. Model 2: age, gender, and race were adjusted. Model 3: age, gender, race, education level, PIR, smoking, alcohol drinking, diabetes, triglycerides, and high-density lipoprotein cholesterol were adjusted.

quartile (P for trend < 0.001). Using the lowest group as a reference, the OR of total periodontitis prevalence for subjects belonging to the second, third, and fourth WWI quartiles were 1.33, 1.65, and 2.10, respectively. In the fully adjusted, participants within the highest quartile of WWI showed a 27% increased likelihood of experiencing periodontitis compared to those in the lowest quartile (OR: 1.27, 95% CI: 1.10–1.46).

### Correlation between WWI and periodontitis severity

As mentioned, we similarly used the same three models to explore the correlation between WWI and different levels of periodontitis severity, including mild, moderate, and severe cases (S2 Table). Across all three models, an elevated WWI was significantly linked to a risen likelihood of moderate periodontitis. Particularly within Model 3, which excluded the influence of all covariates, every unit increment in WWI correlated with a 23% rise in the incidence of moderate periodontitis (OR: 1.23, 95% CI: 1.15–1.32). For mild and severe periodontitis, the increase was comparatively lower at 12% (Mild periodontitis, OR: 1.12, 95% CI: 1.04–1.20; Severe periodontitis, OR: 1.12, 95% CI: 1.01–1.25).

### Non-linear correlation between WWI and periodontitis

The study uncovered a non-linear positive correlation between WWI and periodontitis (Fig 2). Utilizing a two-stage linear regression model, we identified a substantial threshold effect between WWI and periodontitis, with an inflection point (K) calculated at 11.99 (S3 Table).

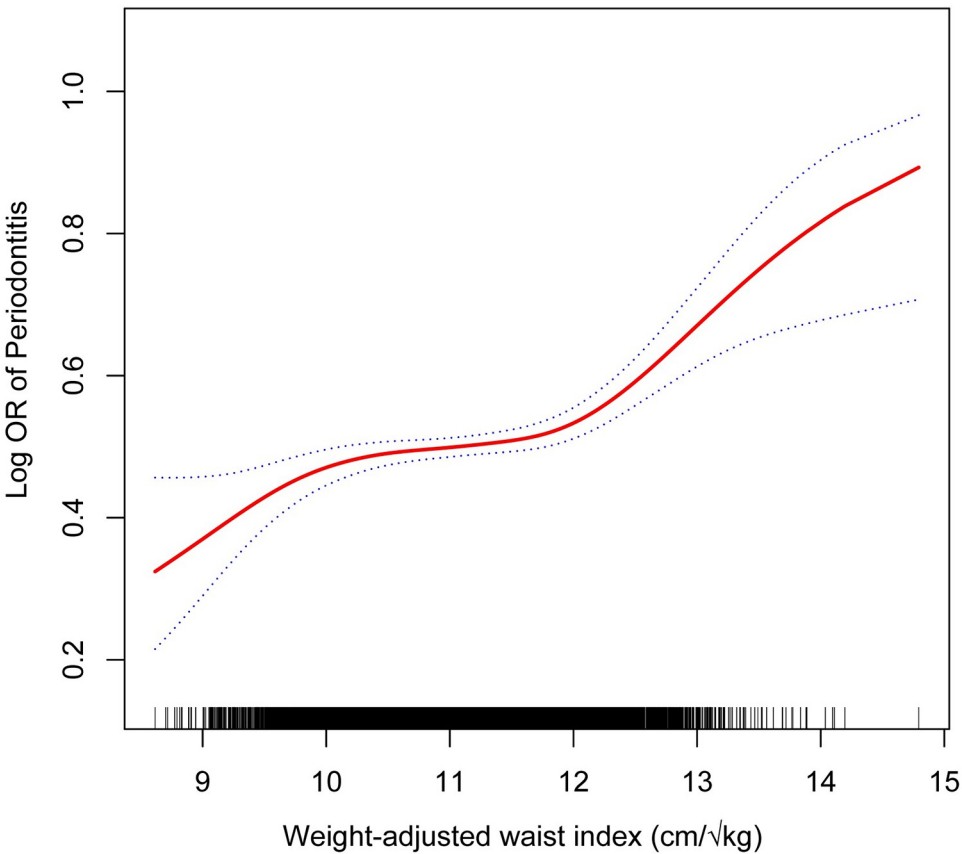

**Fig 2. Non-linear correlation between WWI and periodontitis.**

WWI maintained a significant positive correlation with periodontitis on both sides of this inflection point. For WWI values below the inflection point of 11.99, each unit increase was associated with an 11% rise in periodontitis prevalence (OR: 1.11, 95% CI: 1.03–1.20; P for trend = 0.0096). For WWI values above 11.99, each additional unit was linked to a 97% upsurge in prevalence (OR: 1.97, 95% CI: 1.48–2.62; P for trend < 0.0001). The log-likelihood ratio tests yielded P-values all below 0.001.

## Subgroup analyses

We conducted subgroup analysis and interaction tests to investigate the stability of the relationship between WWI and periodontitis across diverse population backgrounds (Table 3). Our findings revealed significant variations in the WWI-periodontitis association across different age groups (P for interaction = 0.005). Notably, a more robust and pronounced connection was shown in the younger and middle-aged population under 60. For individuals aged 60 and above, each unit's increase in WWI was associated with a 19% rise in periodontitis prevalence. By contrast, in the under-60 age group, this increase was markedly higher, at 44%. None of the other stratified factors such as gender, race, education level, smoking behavior, and diabetes significantly influenced the positive association between WWI and periodontitis (P for interaction > 0.05).

## Discussion

Through this comprehensive cross-sectional study with 10,289 diverse participants, we identified a positive correlation between WWI and periodontitis. Remarkably, age was found to

**Table 3. Subgroup analysis of the association between weight-adjusted waist index and periodontitis.**

| Subgroup | Periodontitis [OR (95%CI)] | P for interaction |
|---|---|---|
| Gender | | 0.4301 |
| Male | 1.17 (1.06, 1.29) 0.0016 | |
| Female | 1.23 (1.12, 1.34) <0.0001 | |
| Age | | 0.0054 |
| < 60 years | 1.44 (1.33, 1.56) <0.0001 | |
| ≥ 60 years | 1.19 (1.06, 1.33) 0.0032 | |
| Race/ethnicity | | 0.1111 |
| Non-Hispanic White | 1.20 (1.09, 1.33) 0.0003 | |
| Non-Hispanic Black | 1.13 (0.98, 1.30) 0.0904 | |
| Mexican American | 1.47 (1.22, 1.77) <0.0001 | |
| Other race/multiracial | 1.13 (0.97, 1.31) 0.1066 | |
| Education level, (%) | | 0.1660 |
| Less than high school | 1.34 (1.16, 1.55) <0.0001 | |
| High school | 1.11 (0.97, 1.28) 0.1326 | |
| More than high school | 1.19 (1.08, 1.30) 0.0003 | |
| Smoked at least 100 cigarettes in life | | 0.7340 |
| Yes | 1.19 (1.08, 1.31) 0.0004 | |
| No | 1.21 (1.11, 1.32) <0.0001 | |
| Diabetes, (%) | | 0.5284 |
| Yes | 1.26 (1.06, 1.50)) 0.0090 | |
| No | 1.19 (1.10, 1.28) <0.0001 | |

Age, gender, race, education level, PIR, smoking, alcohol drinking, diabetes, triglycerides, and high-density lipoprotein cholesterol were adjusted.

have a substantial impact on this correlation, suggesting that higher WWI levels are associated with an increased probability of moderate to severe periodontitis, particularly in individuals under 60 years old. We further discovered that the positive correlation between WWI and periodontitis is nonlinear, with an inflection point at 11.99 cm/$\sqrt{}$kg. This indicates that obesity management assessed by WWI could reduce both the incidence and severity of periodontitis.

Obesity, increasingly recognized as a non-communicable chronic inflammatory disease, is marked by persistently activated multisystemic low-level chronic inflammation and resultant tissue damage. Research indicates that obesity is a primary risk factor for the inflammatory degeneration of periodontal tissues, ranking just behind smoking in its impact [27]. This underlines the potential importance of appropriately evaluating obesity for predicting periodontitis risk and gauging disease severity. Liu et al.'s investigation using NHANES data (2011–2014) found significant correlations between periodontitis and both BMI and WC, though this association was not evident in men [11]. However, Alsalihi et al. and Kim et al. did not find any link between periodontitis and BMI, but an association with WC, whereas Torrungruang et al. reported no association between periodontitis and either BMI or WC [14, 28, 29]. These discrepancies question the effectiveness of these traditional indicators. A comprehensive cohort study appreciated WWI's superiority in predicting cardiometabolic diseases and mortality over BMI, WC, and Waist-to-Height ratio [17]. The WWI was proven to be associated with diabetes, cardiovascular disease, and hypertension, all of which are connected to periodontitis [30–33]. Therefore, we utilized WWI as a new indicator of central obesity reflecting body fat distributions to explore the relationship between obesity and periodontitis. We identified a significant positive link between WWI and a higher prevalence of periodontitis, with a noteworthy nonlinear association and a critical inflection point at 11.99 cm/$\sqrt{}$kg. Above this point, every unit rise in WWI was linked to a 97% increase in periodontitis prevalence. These findings advocate for rigorous WWI management in obesity control as a complementary strategy for improving periodontal health.

Our research further explored the connection between obesity and periodontitis severity, uncovering a novel and significant association: higher WWI levels were notably linked to a greater occurrence of moderate-to-severe periodontitis. This finding contrasts with a Northern Ireland study involving Western European men aged 60 to 70, where BMI was linked to mild periodontitis but not to its severe form [34]. Similarly, another study found no substantial link between BMI and the occurrence of periodontal pockets with PPD exceeding 6 mm [35]. It would be interesting to evaluate the association between WWI and the severity of periodontitis in these populations to see if WWI could reveal a significant correlation. If further research across diverse populations confirms the link between WWI and periodontitis severity, WWI could emerge as a crucial marker for monitoring periodontitis progression and supply evidence for the link between inflammatory disease progression and visceral obesity. Moreover, examining various populations might reveal how racial and regional differences impact this relationship.

The link between WWI and periodontitis was significant across both over and under 60 years groups but was more evident and had a more substantial effect size in the younger cohort. These results diverge somewhat from the previous studies by Liu et al. and Al-Zahrani et al., which reported substantial links between obesity and periodontitis among young and middle-aged adults but not in the older population, using BMI and WC as indicators [11, 12]. On the whole, the correlation between obesity and the incidence of periodontitis appears to be stronger in young and middle-aged individuals compared to older adults. The differences may stem from several factors. Firstly, older individuals may have already lost several teeth, leaving behind healthier remaining teeth. Secondly, oxidative stress and reduced physical activity associated with aging may contribute to increased body fat [36–38]. This implies that weight gain

in some older obese individuals could be attributed to the aging process, whereas obesity in younger age groups might commence earlier in life. Lastly, younger and older populations may adopt different lifestyles, with unhealthy dietary habits prevalent among young people, such as high sugar and high-fat intake, potentially accentuating the relationship between obesity and periodontal disease [39]. Overall, our results endorse WWI's reliability in relation to periodontitis, suggesting a strong correlation between periodontitis and central obesity. However, in our study, neither smoking nor diabetes made any difference. Given the inherent limitation of NHANES dataset to decipher this kind of relationship, there is a need to validate this index on other cohorts longitudinally in the future.

The underlying biological mechanism connecting central obesity WWI to periodontitis remains incompletely understood. Research indicated that adipose tissue secretes various bioactive molecules, such as tumor necrosis factor-$\alpha$ (TNF-$\alpha$), interleukins (IL-6 and IL-8), leptin, and plasminogen activator inhibitor-1 (PAI-1) [40]. Obesity triggers low-grade systemic inflammation through an increase in pro-inflammatory mediators and a decrease in anti-inflammatory factors [41]. This systemic inflammation is also manifested in periodontal tissues, as indicated by elevated concentrations of pro-inflammatory markers (such as TNF-$\alpha$, IL-1, IL-6,) in the gingival crevicular fluid of obese individuals [42]. These inflammatory changes potentially modify the body's immune response, increasing susceptibility to bacterial infections and aggravating the inflammatory and destructive processes of periodontitis induced by oral microorganisms [43]. TNF-$\alpha$, a key cytokine produced in response to periodontal pathogens, plays a crucial role in periodontitis by promoting osteoclast formation and thus contributing to alveolar bone and connective tissue degradation [44]. Moreover, the mechanism of obesity-induced periodontitis might also involve increased insulin resistance [45], heightened oxidative stress [46], changes in gingival blood supply and microcirculation [47], and alterations in oral microbiota composition [48–50]. For example, elevated PAI-1 expression in visceral fat triggers blood coagulation, reducing gingival blood flow, and thus advancing periodontitis progression [47]. Changes in oral microbiota due to obesity are also noteworthy. While Rahman et al. [49] observed a higher presence of periodontitis-associated bacteria like Aggregatibacter actinomycetemcomitans, Tannerella forsythia and Treponema denticola in obese individuals with moderate-severe periodontitis, Lê et al. reported a decreased relative abundance of certain periodontal pathogens in obese subjects. This necessitates further investigation to clarify the disease's pathophysiological mechanisms [50].

Our study stands as a pioneering cross-sectional analysis investigating the relationship between WWI and periodontitis, marked by several strengths. Firstly, the substantial sample size and thorough covariates' adjustment bolster the study's credibility and applicability. Secondly, an essential contribution of this research is identifying the nonlinear and threshold effects in the WWI-periodontitis relationship. However, certain limitations must be acknowledged. The cross-sectional design of our study restricts our capacity to draw causal inferences. Despite integrating numerous relevant covariates into our model, fully mitigating the impact of other potential confounding variables remains challenging. Besides that, the use of the NHANES dataset, specific to the U.S. population, may constrain the global generalization of our findings. Finally, this study considered wide range of demographic data and clinical parameters, but the presence of biochemical parameters may affect the value of this study. While WWI is less commonly used than BMI and WC in obesity assessment, its effectiveness in predicting periodontitis risk and severity suggests the need for further clinical exploration to fully understand its advantages and drawbacks. Despite these constraints, our study demonstrated the relationship between WWI and periodontitis, emphasizing the potential of WWI in evaluating the risks of periodontitis associated with obesity.

## Conclusion

Elevated WWI levels are significantly associated with an increased prevalence of periodontitis. The current findings emphasize the importance of WWI levels in evaluating periodontitis, particularly among the non-elderly population where it may offer enhanced benefits. As WWI is an indicator of adiposity central obesity, our study suggests a tight correlation of central obesity and periodontitis. Nevertheless, validation by further prospective studies is still needed.

## Supporting information

**S1 Table. Checklist of STROBE statement.**
(DOCX)

**S2 Table. The associations between weight-adjusted waist index and severity of periodontitis.**
(DOCX)

**S3 Table. Threshold effect analysis of WWI on periodontitis using two-piecewise linear regression model.**
(DOCX)

**S1 Data.**
(XLS)

## Acknowledgments

The authors thank all of the people who participated in this study.

## Author Contributions

**Conceptualization:** Lihan Xu, Yuntao Liu, Haojing Zhou.

**Data curation:** Lihan Xu, Yuntao Liu.

**Formal analysis:** Lihan Xu, Chenchen Zhao.

**Validation:** Lihan Xu, Haojing Zhou.

**Writing – original draft:** Lihan Xu, Jiaying Wang.

**Writing – review & editing:** Lihan Xu, Yuntao Liu, Chenchen Zhao, Jiaying Wang, Haojing Zhou.

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
