## [Decision Letter · Decision Letter 0]

30 Apr 2024

PONE-D-24-08383Association between weight-adjusted waist index and periodontitis: a population-based studyPLOS ONE

Dear Dr. Zhou,

Thank you for submitting your manuscript to PLOS ONE. After careful consideration, we feel that it has merit but does not fully meet PLOS ONE’s publication criteria as it currently stands. Therefore, we invite you to submit a revised version of the manuscript that addresses the points raised during the review process.

**ACADEMIC EDITOR:**

**I thank you for submitting an interesting topic for publication. However, there are a few queries raised by the reviewers' which if addressed could make your manuscript more robust.**

We look forward to receiving your revised manuscript.

Kind regards,

Tanay Chaubal

Academic Editor

PLOS ONE

Journal Requirements:

Reviewers' comments:

Reviewer's Responses to Questions

**Comments to the Author**

1. Is the manuscript technically sound, and do the data support the conclusions?

Reviewer #1: Yes

Reviewer #2: No

2. Has the statistical analysis been performed appropriately and rigorously? 

Reviewer #1: Yes

Reviewer #2: Yes

3. Have the authors made all data underlying the findings in their manuscript fully available?

Reviewer #1: Yes

Reviewer #2: Yes

4. Is the manuscript presented in an intelligible fashion and written in standard English?

Reviewer #1: Yes

Reviewer #2: Yes

5. Review Comments to the Author

Reviewer #1: The extracted NHANES data along with statistical analysis and the manuscript concluded the study well. Even all 3 models whether covariates not adjusted or some or all covariates adjusted, the relationship of WWI & Periodontitis showed the same pattern. I have certain suggestions to improve this manuscript as follows.

1) The authors should include justification for more pronounced positive correlation in below 60 years group.

2) This study considered wide range of demographic data and clinical parameters, but presence of biochemical parameters could have raised the value of this study (link between obesity and periodontitis). Kindly include it in drawbacks of this study.

3) The WWI is not validated yet and so there is a risk of overfitting and optimism in evaluation of the predictive performance (To fulfill all the 22 checklists of TRIPOD)!

4) Page 9 line 65-66: "Nevertheless, the particular relationship between WWI and periodontitis has rarely been studied comprehensively." If this the first study correlating WWI & Periodontitis, kindly reframe the sentence.

Reviewer #2: the periodontitis classification of the mild moderate and severe were based on what criteria ?any particular article or which year classification was referred?

just the population age was considered or the gender was also taken in account as male and female ? was that also taken or just the age ? will that lead to any limitation to your study when it comes to gender segregation?

6. PLOS authors have the option to publish the peer review history of their article (what does this mean?). If published, this will include your full peer review and any attached files.

Reviewer #1: **Yes: **DR. ANUJ SHARMA

Reviewer #2: No

---

## [Author Response · Author response to Decision Letter 0]

7 May 2024

Reply to Academic Editor

1.Please ensure that your manuscript meets PLOS ONE's style requirements, including those for file naming.

Response: Thank you for your suggestion. We have made the requested changes, including adjusting the font size of the third-level headings, modifying the naming convention of the figures, and adding the figures at the end of the document. If there are any areas where we have not complied with the requirements, please let us know, and we will make the necessary adjustments immediately.

2.Please note that PLOS ONE has specific guidelines on code sharing for submissions in which author-generated code underpins the findings in the manuscript. In these cases, all author-generated code must be made available without restrictions upon publication of the work.

Response: Is your inquiry directed towards the code of the statistical software utilized in this study? The statistical analysis software employed in this study is derived from publicly available sources, as outlined in the methods section. Consequently, there is no specific code necessitating sharing within this article.

3.Please include captions for your Supporting Information files at the end of your manuscript, and update any in-text citations to match accordingly.

Response: A heading for the supplementary information file has been appended to the end of the manuscript.

4.Please review your reference list to ensure that it is complete and correct. If you have cited papers that have been retracted, please include the rationale for doing so in the manuscript text, or remove these references and replace them with relevant current references. Any changes to the reference list should be mentioned in the rebuttal letter that accompanies your revised manuscript. If you need to cite a retracted article, indicate the article’s retracted status in the References list and also include a citation and full reference for the retraction notice.

Response: We have thoroughly reviewed each reference to ensure that none of them have been retracted. In order to improve the manuscript's quality, we have retained the original references and supplemented them with references 18, 24-26, and 36-39.

Reviewer #1: 

1.The authors should include justification for more pronounced positive correlation in below 60 years group.

Response: Thank you for your suggestion. We have thoroughly deliberated on the points you raised. Please see Line 236-246.

2.This study considered wide range of demographic data and clinical parameters, but presence of biochemical parameters could have raised the value of this study (link between obesity and periodontitis). Kindly include it in drawbacks of this study.

Response: Thanks for your advice. We have incorporated the limitation you mentioned. Please see Line 286-288.

3.The WWI is not validated yet and so there is a risk of overfitting and optimism in evaluation of the predictive performance (To fulfill all the 22 checklists of TRIPOD)!

Response: You are correct; adhering to the guidelines for manuscript writing is crucial. Nevertheless, our study is a cross-sectional analysis showcasing the correlation between WWI and periodontitis. We did not utilize WWI for predictive purposes concerning periodontitis. Our methodology aligns with the STROBE statement, elucidated in the Methods section, and we have included the STROBE checklist as Supplementary File S1 Table. Kindly consult section Line 72-73 and Supplementary File S1 Table for further information.

4.Page 9 line 65-66: "Nevertheless, the particular relationship between WWI and periodontitis has rarely been studied comprehensively." If this the first study correlating WWI & Periodontitis, kindly reframe the sentence.

Response: Thank you for your suggestion, and we appreciate you pointing out our error. We have now revised it to state: 'Nevertheless, the correlation between WWI and periodontitis has not been thoroughly investigated.' Please see Line 65-66.

Reviewer #2: 

1.the periodontitis classification of the mild moderate and severe were based on what criteria ?any particular article or which year classification was referred?

Response: Thanks for your careful work. The classification criteria for mild, moderate, and severe periodontitis are established by the consensus of the Centers for Disease Control and Prevention (CDC) and the American Academy of Periodontology, a standard widely adopted by similar studies. We have elaborated on this in the article. Please see Line 92-93 and 101-102.

2.just the population age was considered or the gender was also taken in account as male and female ? was that also taken or just the age ? will that lead to any limitation to your study when it comes to gender segregation?

Response: We accounted for both age and gender in our analysis. Age and gender were incorporated as covariates, and their influence on the outcomes was regulated using statistical methodologies (Table 2). Furthermore, we performed subgroup analyses by stratifying participants based on gender and age to reinforce the relationship between WWI and periodontitis (Table 3). The findings demonstrated the consistent robustness of the association between WWI and periodontitis.

---

## [Decision Letter · Decision Letter 1]

22 May 2024

Association between weight-adjusted waist index and periodontitis: a population-based study

PONE-D-24-08383R1

Dear Dr. Haojing Zhou,

We’re pleased to inform you that your manuscript has been judged scientifically suitable for publication and will be formally accepted for publication once it meets all outstanding technical requirements.

Kind regards,

Tanay Chaubal

Academic Editor

PLOS ONE

Additional Editor Comments (optional):

Reviewers' comments:

Reviewer's Responses to Questions

**Comments to the Author**

1. If the authors have adequately addressed your comments raised in a previous round of review and you feel that this manuscript is now acceptable for publication, you may indicate that here to bypass the “Comments to the Author” section, enter your conflict of interest statement in the “Confidential to Editor” section, and submit your "Accept" recommendation.

Reviewer #1: All comments have been addressed

Reviewer #2: All comments have been addressed

2. Is the manuscript technically sound, and do the data support the conclusions?

Reviewer #1: Yes

Reviewer #2: Yes

3. Has the statistical analysis been performed appropriately and rigorously? 

Reviewer #1: Yes

Reviewer #2: Yes

4. Have the authors made all data underlying the findings in their manuscript fully available?

Reviewer #1: Yes

Reviewer #2: Yes

5. Is the manuscript presented in an intelligible fashion and written in standard English?

Reviewer #1: Yes

Reviewer #2: Yes

6. Review Comments to the Author

Reviewer #1: (No Response)

Reviewer #2: thank you for addressing all the queries. keep up the good work. all the best. but then this study in today s world where junk food is eaten so frequently can be considered as a way to pre determine and predict the periodontal condition

7. PLOS authors have the option to publish the peer review history of their article (what does this mean?). If published, this will include your full peer review and any attached files.

Reviewer #1: **Yes: **Dr. Anuj Sharma

Reviewer #2: No

---

## [Editor Report · Acceptance letter]

27 May 2024

PONE-D-24-08383R1 

PLOS ONE

Dear Dr. Zhou, 

I'm pleased to inform you that your manuscript has been deemed suitable for publication in PLOS ONE. Congratulations! Your manuscript is now being handed over to our production team.

Kind regards, 

on behalf of

Dr. Tanay Chaubal 

Academic Editor

PLOS ONE